# 1T-MoS_2_ Coordinated Bimetal Atoms as Active Centers to Facilitate Hydrogen Generation

**DOI:** 10.3390/ma14154073

**Published:** 2021-07-22

**Authors:** Qiong Peng, Xiaosi Qi, Xiu Gong, Yanli Chen

**Affiliations:** Guizhou Province Key Laboratory for Photoelectrics Technology and Application, College of Physics, Guizhou University, Guiyang 550025, China; xgong@gzu.edu.cn (X.G.); ylchen5@gzu.edu.cn (Y.C.)

**Keywords:** 1T-MoS_2_ monolayer, single metal atom catalysts, dual active centers, hydrogen evolution activity, reaction mechanism

## Abstract

Anchoring single metal atoms has been demonstrated as an effective strategy to boost the catalytic performance of non-noble metal 1T-MoS_2_ towards hydrogen evolution reaction (HER). However, the dual active sites on 1T-MoS_2_ still remain a great challenge. Here, first-principles calculations were performed to systematically investigate the electrocatalytic HER activity of single and dual transition metal (TM) atoms bound to the 1T-MoS_2_ monolayer (TM@1T-MoS_2_). The resulted Ti@1T-MoS_2_ exhibits excellent structural stability, near-thermoneutral adsorption of H* and ultralow reaction barrier (0.15 eV). It is a promising single metal atom catalyst for HER, outperformed the reported Co, Ni and Pd anchoring species. Surprisingly, by further introducing Pd atoms coordinated with S atoms or S vacancies on the Ti@1T-MoS_2_ surface, the resulted catalyst not only maintains the high HER activity of Ti sites, but also achieves new dual active moiety due to the appropriate H* adsorption free energy on Pd sites. This work is of great significance for realizing dual active centers on 1T-MoS_2_ nanosheets and offers new thought for developing high-performance electrocatalysts for HER.

## 1. Introduction

The electrocatalytic hydrogen evolution reaction (HER: 2H^+^ + 2e^−^ → H_2_) is the key to achieve clean hydrogen energy [1,2]. Highly active and stable electrocatalysts are required to drive the sluggish thermodynamic and kinetic processes of HER [3]. Until now, Pt/C is the benchmark catalyst for hydrogen production [4]. Nevertheless, the limited reserve and high price severely restrict its large-scale application. Considering that the HER kinetics under acidic conditions is much faster than that in alkaline solution, thus, it is highly desirable and particularly urgent to mining high activity, low cost, stable electrocatalysts for HER in acidic media [5]. Recently, transition metal embedded heteroatom-doped carbon nanosheets [6], two-dimensional (2D) transition metal sulfides [7,8], borides [9], oxides [10], phosphides [11] etc., have been reported as promising candidates to replace precious Pt for HER. However, so far, these catalysts are still far from being commercialized because of their poor conductivity or insufficient hydrophilicity.

In contrast, the advantages of two-dimensional (2D) 1T phase molybdenum disulfide (1T-MoS_2_) nanosheet are its excellent electrical condutivity, sufficient hydrophilicity, acid resistance and low cost [12]. Compared to the semiconducting 2H phase MoS_2_, the great HER activity of metallic 1T-MoS_2_ mainly originates from its affinity for binding H at the surface S sites, and the improved charge transfer kinetics [13]. Unlike 2H-MoS_2_ where the catalytic activity arises from the edges, the much greater active surface area of 1T nanosheets with respect to the edge portion thus guarantees the higher HER activity [14,15]. Such catalyst has attracted enormous research interests and been regarded as one of the most promising electrocatalysts toward hydrogen evolution in acidic solutions [13,16,17]. Note that a large gap still exists for 1T-MoS_2_ nanosheet to completely substitute Pt because of the high reaction barrier (1.15 eV) during the HER process [13]. In this regard, several efficient approaches have been proposed to boost the catalytic activity of catalysts, such designing novel ferrite-based materials [18], doping TiO_2_ thin films [19] and fabricating Ni Nanowires [20]. In addition, anchoring single transition metal (TM) atoms on the 2D conductive substrates also emerges as an essential method for obtaining superior catalysts [21]. The experimental and theoretical works have reported that single Co, Ni or Pd atoms anchored on the 1T-MoS_2_ basal plane exhibit the Pt-like electrocatalytic activity for HER in acid electrolyte and dramatically improve the durability because of the synergetic effect [16,17,22]. Notably, the strain induced by lattice mismatch and the formation of TM-S covalent bond in TM@1T-MoS_2_ hybrids are favorable for achieving the phase transformation of MoS_2_ from the semiconductive 2H to distorted metallic 1T phase [23]. The appropriate TM atoms integrated into 1T-MoS_2_ efficiently contribute to maintaining the structural stability of 1T phase, also enhancing the corrosion resistance [24]. However, the HER activity of bimetal atoms bound to 1T-MoS_2_ remains to be elucidated.

For mining superior catalysts, it is convenient and efficient to preliminarily evaluate their catalytic activity and stability via integrated computation [25]. In this work, first-principles calculations and automatic frameworks of material screening methods were employed to reveal the electrocatalytic HER activity of 3d, 4d, 5d TM single and dual atoms bound to the 1T-MoS_2_ monolayer. Interestingly, we found that the Ti@1T-MoS_2_ and PdTi@1T-MoS_2_ catalysts exhibit excellent hydrogen evolution activity, comparable to the benchmark Pt catalyst. This work provides significant theoretical insights for the experimental synthesis of superior HER electrocatalysts.

## 2. Materials and Methods

### 2.1. Density Functional Theory Calculations

The structure and catalytic performance of mono/bimetal atoms mediated 1T-MoS_2_ were investigated through first-principles calculations based on the spin-polarized density functional theory (DFT) [26] and automatic frameworks of material screening methods. This process was implemented in the Vienna ab initio simulation package (VASP) [27], combined with the high-throughput computational platform of Artificial Learning and Knowledge Enhanced Materials Informatics Engineering (ALKEMIE) [28]. VASP uses the projector-augmented wave method to describe the electron-core interaction, and allows performing structural optimizations, total-energy calculation, electronic structure calculation and ab initio molecular dynamics simulations [27]. It is currently one of the most popular commercial software in material simulation and computational material science. While, ALKEMIE is an open-source intelligent computational platform for accelerating materials discovery and design via high throughput calculations, data management with the private/shared database, and data mining through machine learning [28]. The exchange-correlation interaction was treated by using the generalized gradient approximation (GGA) in the form of Perdew-Burke-Ernzerhof (PBE) [29]. The 1T-MoS_2_ supercell consisting of 12 Mo and 24 S atoms was used as a model with vacuum layer as large as 20 Å along the z direction. The van der Waals interaction was considered using the empirical correction in the Grimme’s scheme (DFT+D2). The cut-off energy for plane waves was set to 400 eV. While the convergence tolerance was set to 10^−4^ eV in energy and 0.02 eV/Å in force during structural relaxation as well as static calculations.

### 2.2. Formation Energy

To evaluate the thermodynamic stability of mono/bimetal atoms immobilized on the basal plane of 1T-MoS_2_, the formation energy $\Delta E_{f}$ was examined based on Equation (1)
(1)$$\Delta E_{f}=E_{\mathrm{TM}@1T\text{-}\mathrm{Mo}S_{2}}-E_{1T\text{-}\mathrm{Mo}S_{2}}-E_{\mathrm{TM}}^{\mathrm{bulk}}$$
or
(2)$$\Delta E_{f}=E_{\mathrm{TMTi}@1T\text{-}\mathrm{Mo}S_{2}}-E_{1T\text{-}\mathrm{Mo}S_{2}}-E_{\mathrm{TM}}^{\mathrm{bulk}}-E_{\mathrm{Ti}}^{\mathrm{bulk}}$$
where $E_{\mathrm{TM}@1T\text{-}\mathrm{Mo}S_{2}}$ and $E_{\mathrm{TMTi}@1T\text{-}\mathrm{Mo}S_{2}}$ represent the total energies of single TM and bimetal TM/Ti atoms anchored on 1T-MoS_2_, respectively. The terms of $E_{1T\text{-}\mathrm{Mo}S_{2}}$ and $E_{\mathrm{TM}}^{\mathrm{bulk}}$ (or $E_{\mathrm{Ti}}^{\mathrm{bulk}}$) refer to the total energies of the pristine 1T-MoS_2_ and a metal atom in the bulk phase, respectively. Accordingly, the more negative value of $\Delta E_{f}$ indicates the higher thermodynamic stability of the hybrid system.

### 2.3. Reaction Free Energy

For HER, note that this work does not consider the Volmer-Heyrovsky mechanism, because of the large computational cost involved in aqueous solvent models. Instead, the Volmer-Tafel mechanism is used to explore the reaction kinetics of HER, which involves gas phase simulation of H* and H_2_*. Additionally, because the HER kinetics under acidic condition is much faster than that in alkaline solution, electrocatalysts usually show better performance in acidic solution [14]. Therefore, this work mainly aims at the HER performance of mono/bimetal atoms mediated 1T-MoS_2_ in acidic solution. Such HER process consists of two elementary steps via Volmer-Tafel mechanism [24]:(3)$$H^{+}+e^{-}+*\to H*$$
(4)$$2H*\to *+H_{2}(g)$$
where * represents the active center. The Gibbs free energy ∆*G* of H* adsorption was defined as,
(5)$$\Delta G=\Delta E-\Delta E_{\mathrm{ZPE}}-T\Delta S$$
where $\Delta E=E_{{\mathrm{TM}@1T\text{-}\mathrm{MoS}}_{2}+nH}-E_{{\mathrm{TM}@1T\text{-}\mathrm{MoS}}_{2}+(n-1)H}-1/2E_{H_{2}}$ is the adsorption energy needed to increase the coverage by one H atom. $E_{{\mathrm{TM}@1T\text{-}\mathrm{MoS}}_{2}+nH}$ and $E_{{\mathrm{TM}@1T\text{-}\mathrm{MoS}}_{2}+(n-1)H}$ are the total energy of the TM@1T-MoS_2_ hybrids with *n* and *n* − 1 adsorbed H atoms, respectively. And $E_{H_{2}}$ is the total energy of H_2_ molecule. Δ*E*_ZPE_ and Δ*S* represent the change in zero-point energy and entropy of the reaction at 298 K, respectively. In this work, the calculated Δ*E*_ZPE_ + Δ*S* = 0.26 eV, thus, Δ*G*_H_ is rewritten as ΔG=ΔE+0.26 eV. To visually evaluate the catalytic performance, we calculated the theoretical overpotential of HER defined by Equation (5), where low overpotential contributes to accelerating the kinetics process of HER:(6)$$\eta_{\mathrm{HER}}\;=\;|\Delta G_{H*}|/e.$$

## 3. Results

### 3.1. Structural Stability of Single Metal Atoms Mediated 1T-MoS_2_

To rationally design single 3d, 4d and 5d transition metal (TM) atoms immobilized on the 2D 1T-MoS_2_ basal plane (TM@1T-MoS_2_) as HER catalysts, we used the screening procedure as illustrated in Figure 1a. Accordingly, we first constructed 24 TM@1T-MoS_2_ models for structural optimization, then examined their stability by analyzing formation energy, structural evolution, as well as elastic constants, and finally assessed the HER catalytic performance of these stable catalysts by analyzing the free energy of H* intermediate and reaction mechanism. Structural stability is one of the most significant factors to determine the catalytic performance of TM@1T-MoS_2_ materials. For single TM atoms on the basal plane of 1T-MoS_2_, there are three possible adsorption patterns, as shown in Figure 1b. Through structural optimization, the Cr, Mn, Fe, Mo, Ta, W and Re anchored species present obvious structural deformation, indicating these metal atoms adsorbed on 1T-MoS_2_ would be unstable and not suitable for use as a HER catalyst. For other TM atoms adsorption, as reported in previous work of Lau et al. [24], the site c on top of the Mo atom is the most frequently observed adsorbate position with the lowest energy. As two typical representatives, the optimized structures of Ti@1T-MoS_2_ and Pd@1T-MoS_2_ are displayed in Figure 1b, where each TM atom is coordinated with three nonequivalent S atoms. It is worth noting that the adsorption of single TM atoms induces a slight distortion of the MoS_2_ 1T phase into a lower energy 1T′ structure, which, in turn, is stabilized by the adsorbed single atoms. Note that for brevity, we did not distinguish them here. As reported in the previous experimental and theoretical works, anchoring appropriate TM atoms are favorable for maintaining the stability of the 1T metastable phase of MoS_2_ [23,24]. In the stability evaluation step, to specifically examine the thermodynamic stability of TM@1T-MoS_2_ without obvious deformation, the formation energy $\Delta E_{f}$ is calculated, as listed in Figure 1c. The immobilization of 3*d* Ti, V, Co, Ni, Cu, and 4*d* Zr, Nb, Rh, Pd, Ag, as well as 5*d* Hf atoms on 1T-MoS_2_ exhibits very negative formation energy ($\Delta E_{f}$ < −0.85 eV). Particularly, Ti@1T-MoS_2_ has the lowest formation energy, reaching −2.88 eV. Such strong interaction is beneficial to prevent the metal aggregation or being leached [30]. This demonstrates that these 11 TM@1T-MoS_2_ hybrids are thermodynamically stable.

To explore the mechanical stability of thermodynamically stable TM@1T-MoS_2_, their elastic constants are calculated based on the Strain versus Energy method [31,32]. For 2D rectangular TM@1T-MoS_2_ crystals, there are four independent elastic constants, i.e., *C*_11_, *C*_22_, *C*_12_ and *C*_66_. Also, the cases of graphene (*C*_11_ = 357.0 N/m, *C*_12_ = 63.0 N/m) and 2H-MoS_2_ (*C*_11_ = 135.9 N/m, *C*_12_ = 33.3 N/m) are calculated and compared with the previous work [33,34] to verify the reliability of our results. According to the mechanical stability criteria [35], namely, *C*_11_ > 0, *C*_66_ > 0 and *C*_11_ × *C*_22_ > *C*_12_ × *C*_12_, we found that the elastic constants of Zr@1T-MoS_2_ and Nb@1T-MoS_2_ do not meet the criteria. This indicates that the 2D structures of Zr@1T-MoS_2_ and Nb@1T-MoS_2_ are mechanically unstable. Remarkably, for single Ti, V, Co, Ni, Cu, Rh, Pd, Ag and Hf atoms anchored on 1T-MoS_2_, these 9 TM@1T-MoS_2_ catalysts all meet the criteria well (Appendix A), indicating their good mechanical stabilities. For example, the *C*_11_, *C*_22_, *C*_12_ and *C*_66_ of Ti@1T-MoS_2_ are 86.7, 63.3, 46.8 and 9.1 N/m, respectively, which satisfies the mechanical stability criteria well.

Turning our attention to the temperature effect on their stability, ab initio molecular dynamics (AIMD) simulations are performed at and above room temperature (300 and 400 K). Taking Ti@1T-MoS_2_ as an example, the AIMD result in Figure 2a shows that the atoms only slightly vibrate around their equilibrium sites on annealing at 300 K for 10 ps. Relative to the equilibrium state, the deviation of the Ti-S bond length is less than 0.2 Å. A similar case is also found at 400 K (Figure 2b). These results indicate that the Ti@1T-MoS_2_ catalysts are capable of remaining stable at room temperature or even higher. Considering the fact that the catalytic reactions generally occur in an aqueous environment, AIMD approach is also carried out to examine the stability of Ti@1T-MoS_2_ in water solutions, which can be reflected by the fluctuations of the current density in experiment [36]. As shown in Figure 2c, after annealing at a temperature of 300 K for 10 ps, the structure of Ti@1T-MoS_2_ remains intact in water without any noticeable deformation. This fact can be further verified by the uniform evolution of temperature and energy with the simulation time (Figure 2d), where their fluctuations are relatively small. Such findings imply the outstanding stability of Ti@1T-MoS_2_ in aqueous solutions at ambient conditions. In their experiment, Lau et al. [24] provided direct experimental evidence that with proper treatment at environmental conditions, anchoring TM single-atoms on the 1T-MoS_2_ basal plane is feasible and greatly helpful to prevent the reconstruction of MoS_2_ nanosheets from the distorted 1T phase to 2H phase, thereby maintaining the stability of the 1T structure. Overall, the latest experimental progresses in Co@1T-MoS_2_ [23], Ni@1T-MoS_2_ [16], and Pd@1T-MoS_2_ [24] make these TM@1T-MoS_2_ nanomaterials with good thermodynamic, mechanical and thermal stability be promisingly synthesized for task-specific applications in the near future.

### 3.2. HER Activity and Kinetics of TM@1T-MoS_2_

Given the structures of 9 stable TM@1T-MoS_2_ crystals, in-depth study of their HER activity is essential to provide new research ideas for the further development of MoS_2_-based hydrogen production catalysts. In this regard, the adsorption behaviors of H atom are examined at different sites, as marked in Appendix A. According to the calculated total energy listed in Appendix A, the energetically most favorable configurations of H adsorption vary depending on the type of TM atoms fixed on the surface. For Pd@1T-MoS_2_ catalyst, our work demonstrates that S atoms near Pd become activated. Since the used 1T-MoS_2_ supercell contains 12 nonmetal S atoms on the top atomic layer, the coverage (θ) of one H atom adsorption is calculated as 1/12. At this low θ, the adsorbed H atom preferentially binds to S atom on the Pd@1T-MoS_2_ surface, which can be attributed to the fact that the electronegativity difference between H (2.10) and S atoms (2.58) is greater than that of H and Pd atoms (2.20). The corresponding free energy of H* adsorption (Δ*G*_H*_) reaches 0.36 eV, consistent with the reported results (0.35 eV) of Lau et al. [24] In their experiment, they have successfully synthesized Pd@1T-MoS_2_ catalyst and confirmed their high activity for HER to dramatically accelerate the rate limiting recombination of H* to H_2_. In contrast, for Ti@1T-MoS_2_, and V@1T-MoS_2_ systems, it is found that the adsorbed H atom prefers to be directly bonded with the single TM atoms fixed on the basal plane. Therefore, the anchored Ti and V single atoms are identified as the active sites for HER. Generally, the free energy of H* adsorption Δ*G*_H*_ is used as an effective descriptor to evaluate the catalysts’ activity for hydrogen evolution [37]. Too negative or too positive ∆*G*_H*_ will result in slow HER kinetics. Therefore, when Δ*G*_H*_ is as close to zero as possible, the catalyst has the optimal catalytic activity toward HER [38]. It is worth noting that, for Ti@1T-MoS_2_ and V@1T-MoS_2_ catalysts as summarized in Appendix A, their Δ*G*_H*_ is as low as 0.12 and 0.13 eV, respectively. Correspondingly, the absolute value |Δ*G*_H*_| is significantly lower than the reported Pd@1T-MoS_2_ catalyst (0.35) [24], and even comparable to Pt(111) [4] (theoretical Δ*G*_H*_ = −0.09 eV at H coverage of 1/4). A volcano plot composed of Δ*G*_H*_ and overpotential $\eta_{\mathrm{HER}}$ is used to intuitively describe the HER activity of the screened 9 stable TM@1T-MoS_2_ nanomaterials in Figure 3. The closer to the top of the volcano plot, the higher the hydrogen evolution activity of such catalysts [39]. Notably, for the Ti@1T-MoS_2_ and V@1T-MoS_2_ catalysts with acid corrosion resistance, they are closely located at the top of HER volcano plot, reaching 0.12, and 0.13 V, respectively. The ultralow overpotential of single Ti and V atoms anchored on 1T-MoS_2_, comparable to the benchmark Pt(111) ($\eta_{\mathrm{HER}}$ = 0.09 V) [4] and pristine 1T-MoS_2_ (0.14 V) catalysts [13], implies their potential catalytic activity for hydrogen evolution.

To gain insights into the HER kinetics, we took Ti@1T-MoS_2_ as a representative to search for the minimum energy pathway and calculated the reaction barrier through the climbing-image nudged elastic band (NEB) method [40]. As reported in previous work [24], for Pd@1T-MoS_2_ catalyst, the H atom is first adsorbed on the activated S atom and then transferred to the proximal Pd atom to generate H_2_ (Figure 4a). Surprisingly, for the Ti@1T-MoS_2_ catalyst, the first H atom prefers to be directly bonded with the Ti atom anchored on the basal plane, which can effectively boost the efficiency of proton transfer. Then, H* reacts with another adsorbed H* to form H_2_, as displayed in Figure 4b. The obtained results show that the single Ti atom on Ti@1T-MoS_2_ is the principal active center for catalyzing hydrogen evolution. The corresponding minimum energy pathway of Volmer step and Tafel H-H recombination is visually presented in Figure 4c. It is clear that for Ti@1T-MoS_2_ electrocatalyst, the reaction barrier of Tafel step reduces from 1.15 eV (undoped 1T-MoS_2_) [13] to 0.15 eV, even showing much faster reaction kinetics than the reported Pd@1T-MoS_2_ catalyst (0.33 eV) [24]. The anchored Ti atom dramatically accelerates the recombination of H* to H_2_. Therefore, the formed Ti@1T-MoS_2_ hybrid is identified as a superior electrocatalyst toward HER at low H coverage of 1/12.

Furthermore, to demonstrate the catalytic capability of Ti@1T-MoS_2_ at different levels of H* coverage, we constructed the adsorption configuration with increased H* coverage (θ = 1/12, 2/12, 3/12 and 4/12) and calculated their Δ*G*_H*_. Herein, one H atom adsorption corresponds to a surface coverage of 1/12, while the adsorption of four H atoms corresponds to a coverage of 4/12. It is clear that in Appendix A, as the H coverage θ increases from 1/12 to 4/12, the Δ*G*_H*_ of Ti@1T-MoS_2_ becomes more positive due to the lack of electrons in the surface S atoms and the insufficient active sites. Therefore, creating more active sites for the adsorption and activation of the reactant H* is highly significant, in order to achieve high HER activity of the catalyst in a wider H coverage.

### 3.3. Bimetal Active Sites on TMTi@1T-MoS_2_

At present, most studies on the atomic-level active sites of 1T-MoS_2_ only consider one type of single metal active center. Yet, the bimetal active centers on 1T-MoS_2_ remain a great challenge to be solved urgently. When two or more types of single metal atoms are introduced, one metal atom, not only serves as an active site, but also acts as a catalytic accelerator by modifying the electronic structure and coordination environment of another metal atom [41]. Ideally, such hybrids are beneficial to balance the interaction between the active center and reaction intermediate H*, thus, further enhancing the overall HER activity [42]. In view of this, we rationally designed the bimetal active centers on the 1T-MoS_2_ basal plane, and explored the regulation of their HER activity by controlling the synergistic effect between different single metal atoms and 1T-MoS_2_ conductive substrate. As illustrated in the inset of Figure 5a, by further introducing single Ti, V, Mo, Ni, Cu, Pd and Pt atoms into the Ti@1T-MoS_2_ nanomaterial, a new bimetal site was created. After structural optimization, the anchored bimetal atoms also induce a slight structural distortion of 1T-MoS_2_. To confirm their stability, we calculated the formation energy as shown in Figure 5a. Importantly, the TiTi@1T-MoS_2_, VTi@1T-MoS_2_, NiTi@1T-MoS_2_, CuTi@1T-MoS_2_, PdTi@1T-MoS_2_ hybrids show very negative formation energy of lower than −1 eV, implying their good thermodynamic stability. Turning our attention to their hydrogen evolution activity, Figure 5b shows the considered H* adsorption models in which one H atom binds to a metal or nonmetal atom. Here, the PdTi@1T-MoS_2_ catalyst with bimetal sites is taken as a typical representative of these stable TMTi@1T-MoS_2_ hybrids. Interestingly, the H* adsorption properties on the aforementioned five bimetal catalysts are very similar, i.e., the first H atom is inclined to be directly bonded to the electron-rich Ti atom, as summarized in Appendix A. The proximal metal atom plays an efficient role in further activating the Ti atom. Importantly, for VTi@1T-MoS_2_ and CuTi@1T-MoS_2_, their Δ*G*_H*_ of one H atom adsorbed on Ti site is only about −0.01 and +0.01 eV (Figure 5c), respectively, even closer to zero than that on Pt(111) with −0.09 eV. However, the H* adsorption on the V or Cu site shows high Δ*G*_H*_, reaching 0.53, and 0.64 eV (Appendix A), respectively. That is to say, in such catalysts, only one metal site exhibits catalytic activity for HER. Unexpectedly, for PdTi@1T-MoS_2_ catalyst, the free energy Δ*G*_H*_ of one H adsorption on the Pd or Ti sites has very close values, being 0.20, and 0.12 eV, respectively. When H coverage increases from 1/12 to 2/12 (i.e., H* is adsorbed on the Ti and Pd sites simultaneously), the Pd site still maintains the HER activity (Δ*G*_H*_ = 0.20 eV in Appendix A). The bonding strength of the adsorbed H* to the dual Pd and Ti sites is neither too strong nor too weak. Both of them are able to serve as the active sites for H adsorption. For electrocatalyzing hydrogen evolution, the unique dual active centers make PdTi@1T-MoS_2_ applicable for wider H coverage.

### 3.4. Sulfur Vacancy-Mediated Hydrogen Adsorption on TMTi@1T-MoS_2_

Based on the experimental findings of Pető et al., the formation of S vacancy is unavoidable on the basal plane of 2D 1T-MoS_2_ crystals at room temperature [43]. To this end, we investigated the effects of S vacancy on the catalytic activity of bimetal catalysts (denoted as TM_Sv_Ti@1T-MoS_2_, TM = Ti, V, Ni, Cu, Pd and Pt), where Ti atom was coordinated with three S atoms (site-S) and the other TM was deposited into the S vacancy (site-Sv). The reason why the early transition metal Ti was chosen to coordinate with S is that it has a relatively large electronegativity difference with S, favoring the stability of Ti atom immobilized on the 1T-MoS_2_ basal plane. A widely used parameter for predicting the HER activity of specific catalyst is the H adsorption Gibbs free energy (Δ*G*_H*_). We calculated Δ*G*_H_ for H adsorption on diverse sites and compared it to that of pristine TMTi@1T-MoS_2_ without S vacancy. The main findings were presented in Figure 6 and Appendix A. Our theoretical result reveal that for V_Sv_Ti@1T-MoS_2_, Ni_Sv_Ti@1T-MoS_2_, Cu_Sv_Ti@1T-MoS_2_, Pd_Sv_Ti@1T-MoS_2_, Pt_Sv_Ti@1T-MoS_2_ bimetal catalysts, the H atoms with coverage θ = 1/12 prefers to bind to the TM_Sv_ atoms deposited in the S vacancies. Among them, Δ*G*_H*_ of V_Sv_Ti@1T-MoS_2_ displays similar values of 0.18 eV and 0.15 eV at the site-S and site-Sv, respectively (Figure 6b). This means that both V and Ti bimetals of V_Sv_Ti@1T-MoS_2_ are able to act as active sites to boost the hydrogen adsorption and desorption at low H coverage, further enhancing the HER activity. However, at higher H coverage θ = 2/12, Δ*G*_H*_ of V_Sv_Ti@1T-MoS_2_ becomes more positive reaching 0.34 eV, indicating a not very favorable H adsorption. Therefore, the V_Sv_Ti@1T-MoS_2_ catalyst is only suitable for low H coverage.

Differently, Δ*G*_H*_ of Pd_Sv_Ti@1T-MoS_2_ is lowered to 0.09 eV and −0.05 eV at low (θ = 1/12) and higher (θ = 2/12) H coverage, respectively (see Figure 6d,f). These findings clearly show that at low H coverage, the Pd atoms deposited on the S vacancies (Pd_Sv_) serve as the main active sites for H adsorption. While at high H coverage, both Pd_Sv_ and Ti acts as the dual active sites to facilitate the hydrogen evolution. The enhanced catalytic activity is clearly related to the presence of substitutional Pd_Sv_ sites, which saturate the S vacancies and induce charge redistribution. The experimental results of Lau et al. [24] also confirmed the enhanced HER activity of 1T-MoS_2_ monolayer doped with Pd atom at the S vacancy site.

## 4. Discussion

To deeply gain insight into the underlying modulation mechanism for PdTi@1T-MoS_2_ catalyst, we investigated its electronic structure as illustrated in Figure 7. Introducing single Pd atom into Ti@1T-MoS_2_ can alter the coordination environment of the initially anchored Ti site, leading to the rearrangement of *d* orbitals. From the projected density of states shown in Figure 7a, it is clear that the electron-rich Ti-*d* orbital mainly occupies the conduction band above the Fermi level, ranging from 0.5 to 1.5 eV. Differently, the Pd-*d* orbital mainly dominates the valence band ranging from −4 to −1 eV. As a result, the PdTi@1T-MoS_2_ hybrid shows excellent electronic conductivity, which, in turn, boosts the catalytic activity for hydrogen production. In addition, the electron localization function (ELF) [44] is calculated to characterize the localized distribution characteristics of electrons, involving atomic binding and lone electron pairs in the PdTi@1T-MoS_2_ system. It is displayed in Figure 7b, where the upper limit 1 of ELF corresponds to the perfect localization of electrons, while the lower limit 0 for complete delocalization (or no electrons), and the middle value 0.5 corresponds to electron gas-like pair probability. It is clearly seen that for single Ti and Pd atoms co-fixed on the 1T-MoS_2_ crystal, the red area is mainly distributed outside the S atom, where ELF value reaches 0.77~0.82. The highly localized electrons of S atom are not favorable for the adsorption of the reactant H*. Notably, the area near the Pd and Ti atoms is shown in light blue, and the corresponding ELF value is 0.13 or even lower. This indicates that the electrons around Ti and Pd are highly delocalized, thus contributing to the adsorption of proton hydrogen, further achieving more catalytic active sites. These results confirm the superior HER performance of PdTi@1T-MoS_2_ electrocatalyst due to its improved active center and good conductivity.

Our theoretical findings provide an important guidance for the experimental synthesis of target HER catalyst. By introducing monodisperse Pd and Ti atoms on the basal plane, the precise engineering atomic structures contributes to the formation of more active sites on 1T-MoS_2_. Several powerful synthetic methods have been proposed to precisely fabricate the atomic dispersed catalysts, including mass-selected soft landing, atomic layer deposition (ALD), co-precipitation and impregnation wet-chemical routes [45,46]. Therefore, we hope that the PdTi@1T-MoS_2_ catalyst can be fabricated in the near future for electrocatalyzing HER in acid electrolytes.

## 5. Conclusions

In summary, we reported the mono/bimetal atoms mediated 1T-MoS_2_ as highly stable and efficient electrocatalysts for HER via systematically controlling the composition and electronic structure. Using the density functional theory and automatic frameworks of material screening methods, it is found that the single Ti atoms anchored on the 1T-MoS_2_ (Ti@1T-MoS_2_) basal plane are able to serve as the active center for H* adsorption instead of the proximal S. The obtained Ti@1T-MoS_2_ catalyst exhibits excellent structural stability, near-thermoneutral adsorption of H* and ultralow reaction barrier (0.15 eV), outperformed the reported Co, Ni, Pd anchoring species. More importantly, for bimetal Pd and Ti atoms immobilized together on 1T-MoS_2_, the major advantage of such PdTi@1T-MoS_2_ catalyst is that, it maintains the high activity of single Ti atom, and achieves new dual active sites, where the bonding strength of adsorbed H* to the dual Pd and Ti sites is neither too strong nor too weak. The unique dual active centers make PdTi@1T-MoS_2_ applicable for wider H coverage, comparable to the benchmark Pt catalyst. This work provides an attractive design strategy to improve the electrocatalytic activity of 1T-MoS_2_ for HER.

## Figures and Tables

**Figure 1 materials-14-04073-f001:**
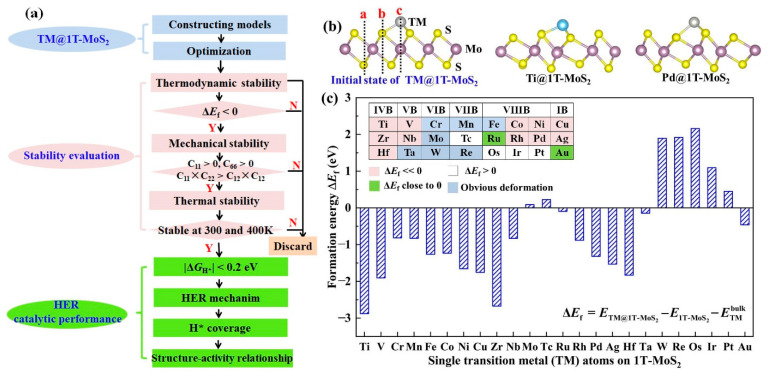
(**a**) Screening procedures of single transition metal (TM) atoms anchored on the monolayer 1T-MoS_2_ basal plane (TM@1T-MoS_2_) as HER electrocatalysts. (**b**) Initial and optimized structure of Ti@1T-MoS_2_ and Pd@1T-MoS_2_, where the a, b, c sites represent the possible initial adsorption patterns of single atoms. (**c**) Formation energy of TM@1T-MoS_2_ hybrids.

**Figure 2 materials-14-04073-f002:**
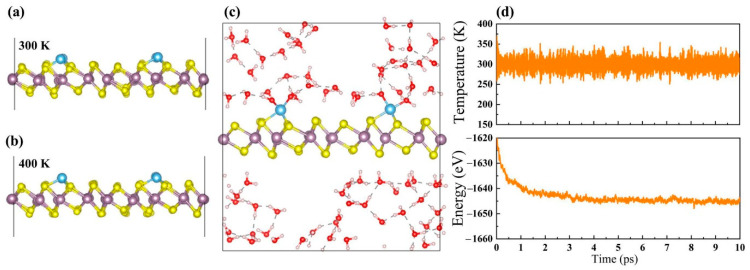
Structures of Ti@1T-MoS_2_ after AIMD simulation (**a**) at 300 K and (**b**) 400 K for 10 ps. (**c**) Structure of Ti@1T-MoS_2_ in water solutions after AIMD simulation at 300 K for 10 ps, and (**d**) corresponding evolution of temperature and energy with the simulation time.

**Figure 3 materials-14-04073-f003:**
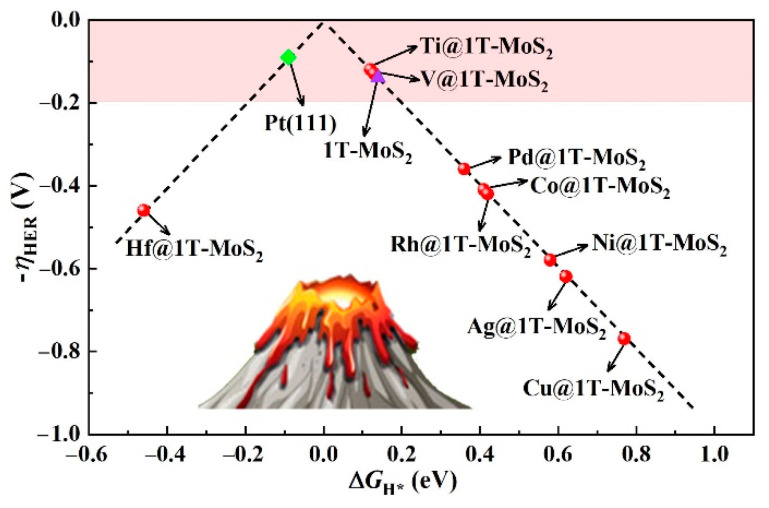
HER volcano plot of TM@1T-MoS_2_ compared to Pt(111) and pristine 1T-MoS_2_ catalysts.

**Figure 4 materials-14-04073-f004:**
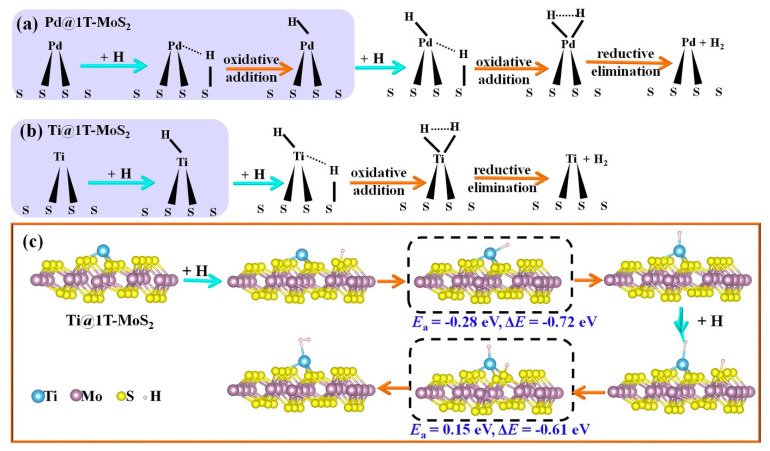
(**a**) Schematic of HER mechanism on Pd@1T-MoS_2_. (**b**) Schematic of HER mechanism on Ti@1T-MoS_2_ and (**c**) the corresponding minimum-energy pathways via Volmer-Tafel route. The transition state is marked with a dashed frame. The activation energy barrier (*E*_a_) and reaction energy (Δ*E*, total energy change between product and reactant) are shown below the dashed frame.

**Figure 5 materials-14-04073-f005:**
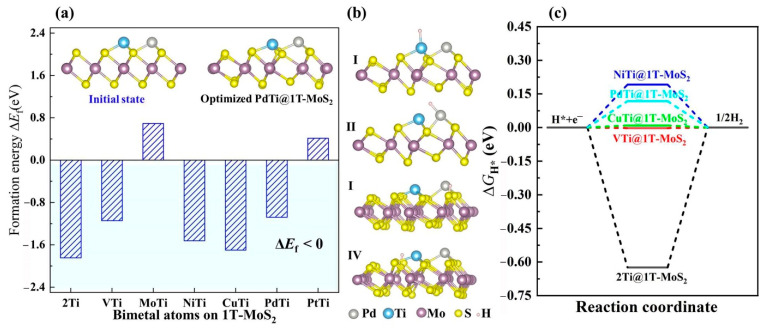
(**a**) Formation energy of TMTi@1T-MoS_2_ hybrids by anchoring bimetal atoms on the 1T-MoS_2_ surface, where the insets show the initial state and optimized equilibrium structure of PdTi@1T-MoS_2_ as an example. (**b**) Side views of the considered H* adsorption sites on PdTi@1T-MoS_2_. (**c**) Δ*G*_H*_ of H* on the lowest energy state for a given bimetal catalyst.

**Figure 6 materials-14-04073-f006:**
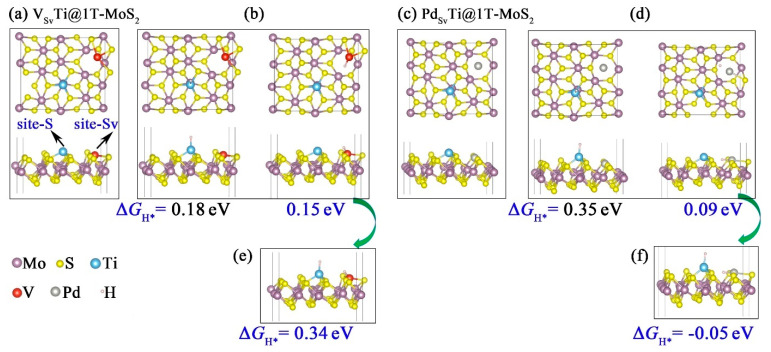
Top and side views of optimized (**a**) V_Sv_Ti@1T-MoS_2_ and (**b**) one H adsorption on V_Sv_Ti@1T-MoS_2_. Top and side views of optimized (**c**) Pd_Sv_Ti@1T-MoS_2_ and (**d**) one H adsorption on Pd_Sv_Ti@1T-MoS_2_. (**e**) Two H adsorption on V_Sv_Ti@1T-MoS_2_. (**f**) Two H adsorption on Pd_Sv_Ti@1T-MoS_2_.

**Figure 7 materials-14-04073-f007:**
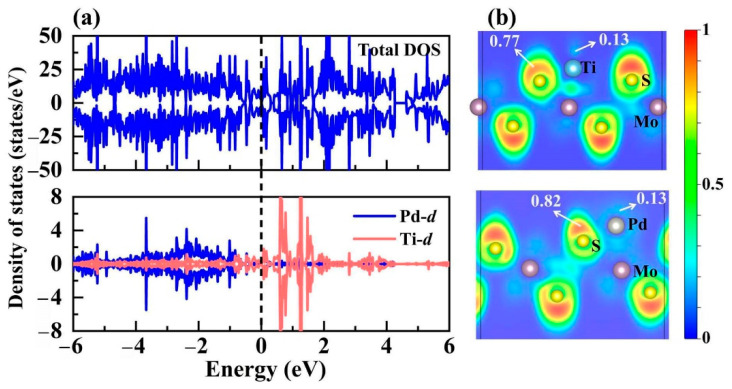
(**a**) Total and projected density of states for PdTi@1T-MoS_2_, as well as (**b**) the electron localization functions (ELF). The Fermi level is set to 0 eV.

## Data Availability

The data presented in this study are available on request from the corresponding author.

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
