# Peer review of "1T-MoS2 Coordinated Bimetal Atoms as Active Centers to Facilitate Hydrogen Generation"

_materials, 2021, doi:10.3390/ma14154073_

Round 1
Reviewer 1 Report
The work presented in the manuscript is novel, interesting and was carefully performed. The implications of the study are apparent.
The reviewer has the following recommendations to the authors:
A. While the original sources of methodologies and tools are adequately cited, including a brief description of each, e.g., VASP or ALKEMIE, would be useful to a broader readership.
B. English style needs to be carefully revised in order to convey accurate information regarding the work. For example, the following sentence in the abstract: "Herein, we designed mono-bimetal active centers on the experimentally available 1T-MoS2 monolayer and revealed their electrocatalytic activity origin for HER based on the first-principles calculations" should be rephrased to better convey the fact that ab-initio simulations were performed, not design per se.
C. English sentence structure revisions would facilitate the presentation of ideas in a straight-forward fashion. For example, this sentence on the abstract can benefit from revisions: "For single Ti atoms anchored on 1T-MoS2 to form Ti@1T-MoS2, such hybrid catalyst exhibits excellent structural stability, near-thermoneutral adsorption of H* and ultralow reaction barrier (0.15 eV), outperformed the reported Co, Ni and Pd anchoring species".
Reviewer 2 Report
The manuscript “1T-MoS2 coordinated bimetal atoms as active centers to facilitate hydrogen generation” is a well readable and well-followed work that deals with very topical issues. The role of water splitting is dominant in energy storage in the XXI. century. MoS2 can play a key role in producing electrodes for hydrogen evolution using as small amount metal as possible.
The authors determined by DFT calculation how the various transition metals and Pd bind in the surface of MoS2 forming single atomic catalytic centers. A rather good suggestion is the study of bimetallic catalysts, well known in classical catalysis, in this case with two different metal atoms. The authors count with a 2D, perfect 1T-MoS2 layer and unfortunately ignore the fact that there are more or less S vacancies on the surface of 2D MoS2 [PetÅ‘, J., et al. Spontaneous doping of the basal plane of MoS2 single layers through oxygen substitution under ambient conditions. Nature Chem 10, 1246–1251 (2018)], which may play a prominent role in the adsorption of transition metal atoms. Lau et al., cited by the authors, [20] also take into account S-vacancy sites during surface bonding of the metal atom. There is probably not much difference in the stability of the metal binding of the catalyst when the metal is adsorbed to the S atom or at the S vacancy site, but there may be a difference in catalytic activity. This difference may be even more significant for bimetal catalysts, as the metal atom that binds to the S vacancy and the one that binds to the S atoms may be in an energetically different position.
Adsorption at S vacancy sites, especially in the case of bimetallic centers, would greatly increase the scientific interest of the manuscript. Therefore, I suggest performing and presenting these calculations. I would consider a manuscript supplemented by these calculations to be acceptable.
Reviewer 3 Report
REFEREE REPORT
on paper “1T-MoS2 coordinated bimetal atoms as active centers to facili-tate hydrogen generation”
by authors Qiong Peng, Xiaosi Qi, Xiu Gong and Yanli Chen,
submitted to Materials
The paper “1T-MoS2 coordinated bimetal atoms as active centers to facili-tate hydrogen generation” is devoted to modelling of the mono/bimetal active centers on the experimentally available 1T-MoS2 monolayer and revealed their electrocatalytic activity origin for hydrogen evolution reaction (HER) based on the first-principles calculations. The topic of this paper is critically actual, especially for realizing dual active centers on 1T-MoS2 nanosheets and offers new thought for developing high-performance HER electrocatalysts. The data are reliable and do not cause much doubt. Nevertheless, there are several points before the paper can be published. I hope that authors after minor revisions can improve the paper and can publish it in Materials.
- The Introduction part must be improved with new literature in the field of the new relevant materials for the HER electrocatalysts and I suggest to use the following reference (see and discuss: https://doi.org/10.1016/j.matchemphys.2021.124818; https://doi.org/10.1007/s10854-020-04626-7; https://doi.org/10.4028/www.scientific.net/MSF.946.235).
- Why the molybdenum disulfide does perform the great HER activity?
- Why did you choose this material as experimental object? Please include the advantages of such material in comparison with traditionally used.
- Discussion part: the authors concluded that presented results provide an attractive design strategy to improve the electrocatalytic activity of 1T-MoS2 for HER. It will be better include the specific suggestions for a strategy for using these materials for HER (more practical applications are needed).
- Authors must improve language. There are some insufficient typos and mistakes in the text.
Round 2
Reviewer 2 Report
Great work!